:ፘ: PLOS | ONE

**Data Availability Statement:** Data for SEER database is readily available at https://seer.cancer.gov/. Instructions for access to dataset are

# Survival of infants ≤24 months of age with brain tumors: A population-based study using the SEER database

Claire Faltermeier[1], Timothy Chai[2], Sharjeel Syed[2], Nathan Lau[3], Lior Elkaim[4], George Ibrahim[5], Anthony Wang[6], Alexander Weil[4], Anne Bendel[7], Aria Fallah[6], Albert Tu [8]*

**1** Department of Neurosurgery, University of California, San Francisco, CA, United States of America, **2** Department of Medicine, Stanford University School of Medicine, Stanford, CA, United States of America, **3** Fred Hutchinson Cancer Research Center, Seattle, WA, United States of America, **4** Division of Neurosurgery, CHU Sainte-Justine Hospital, Université de Montréal, QC, Canada, **5** Division of Neurosurgery, Department of Surgery, Hospital for Sick Children, University of Toronto, ON, Canada, **6** Department of Neurosurgery, Mattel Children's Hospital, University of California, Los Angeles, CA, United States of America, **7** Department of Hematology-Oncology, Children's Minnesota, Minneapolis, MN, United States of America, **8** Division of Neurosurgery, Children's Hospital of Eastern Ontario, ON, Canada

\* Atu@cheo.on.ca

## Abstract

### Introduction

Brain tumors are the most common solid malignancy and leading cause of cancer-related deaths in infants. Current epidemiological data is limited by low numbers of reported cases. This study used a population-based approach with analysis of contemporary and historical survival curves to provide up-to-date prognostication.

### Methods

Observational cohort analysis was performed using the Surveillance, Epidemiology and End Results (SEER) database. Infants with brain tumors diagnosed from 1973 to 2013 were categorized by the most common tumor types (diffuse astrocytic and oligodendroglioma, choroid plexus, embryonal, ependymal, medulloblastoma and pilocytic astrocytoma). The 1, 5 and 10 year survival was stratified by decade, with trends in management and outcomes analyzed.

### Results

We identified 2996 affected infants satisfying inclusion criteria. All tumor types, except embryonal and choroid plexus, demonstrated improving survival with time. Infants with embryonal tumors showed a decline in survival from the 1970s to 1990s (p = 0.009), whereas infants with choroid plexus tumors had no change in survival. Infants with ependymal tumors experienced the greatest improvement in survival from 1980s to 1990s and 1990s to 2000s (p = 0.0001, p = 0.01), with 5-year survival probability improving from 28% (95% CI 15–42%) in the 1980s to 77% (95% CI 69–83%) the 2000s. The use of radiation declined from 1970 to 2000 for all tumors; however, radiation treatment for embryonal and ependymal subtypes increased after 2000.

available on website. All software for data analysis available at https://seer.cancer.gov/.

**Funding:** The authors received no specific funding for this work.

**Competing interests:** The authors have declared that no competing interests exist.

## Conclusions

While overall survival for infants with brain tumors has improved from the 1970s onwards, not every tumor type has seen a statistically significant change. Given changes in management and survival, prognostication of infants with brain tumor should be updated.

## Introduction

Brain tumors are the most common solid malignancy and leading cause of cancer related deaths in patients <24 months of age[1]. Brain tumors in infants differ from tumors in their older counterparts by histological type, management and prognosis[2]. Historically, overall survival of infants with brain tumors is worse compared to their older counterparts[3]. Poor prognosis has been associated with numerous factors including: aggressive biology, presentation at an advanced stage, and treatment-associated neurotoxicity[3].

### Current state of knowledge

The current epidemiological data regarding infantile brain tumors is derived from case series and selected population based studies[4–6]. The limitations of the available data include low numbers and grouping of tumors that does not stratify by histological diagnosis [4, 7, 8]. Furthermore, prior studies used data collected between the 1970s-1980s or combined data from 1970s-2000s for prognostication[9–12]. These studies do not reflect how contemporary management approaches have influenced brain tumor survival.

### Rationale

The rationale for this study is to use a population-based approach applying SEER data to provide contemporary prognostication for infantile brain tumors. This large cancer registry enables analysis of rare tumors such as infantile brain tumors. Data collection began in 1973, making it an ideal source to evaluate cancer survival trends over time[13].

## Methods

### Study design

We performed an observational cohort study with information extracted from the US National Cancer Institute's SEER database. To extract the data, SEER-STAT software version 8.3.4 (National Cancer Institute, Bethesda, MD, USA) was used, as previously described[14]. Analysis included data from SEER's state and county registries from 1973–2013. Current registries in the SEER database include: 8 state registries (Connecticut, Georgia, Hawaii, Iowa, Kentucky, Louisiana, New Mexico and Utah), and multi-county areas of Atlanta, rural Georgia, Detroit, San-Francisco-Oakland, Seattle-Puget Sound, San Jose-Monterey, and Los Angeles. Registries of Alaskan natives, American Indians in Arizona and Alaska, and the Cherokee Nation were also included.

We included all participants aged 24 months and under diagnosed with a brain tumor between the years of 1973–2013. The objectives of the study were to evaluate: 1) the distribution of infantile brain tumors, 2) survival trends for brain tumors by subtype and by epoch, and 3) trends in treatment and shifts in therapeutic approaches over time. The variables collected included: year of diagnosis, age at diagnosis, year of birth, survival months, vital status (categorized as Dead or Alive), cause of death, ICD-10 code, and treatment received. The primary outcome was survival months. The duration of follow-up was determined from the year

of diagnosis, survival months and vital status. The year of diagnosis was used to categorize patients by decade of diagnosis (i.e. 1980–1989, 1990–1999, etc.). The cause of death was categorized as CNS-dependent and CNS-independent. Treatment options were: only radiation; only surgery; both; or neither. ICD-10 codes were used to separate participants into brain tumor histological subtypes. Histological subtypes with fewer than 10 patients were removed from analysis. Remaining subtypes were grouped into 6 major categories: 1) diffuse astrocytic & oligodendroglial, 2) embryonal, 3) ependymal, 4) medulloblastoma, 5) pilocytic astrocytoma and 6) choroid plexus tumors. Kaplan-Meier analyses were generated for all tumor types combined and individually based on decade of diagnosis.

Patients without information on survival months, vital status, or year of diagnosis were excluded from survival analyses. Patients with loss of follow-up and CNS-independent causes of deaths (i.e. cardiac diseases) were censored in all survival analyses. All data parsing was conducted using the software Python version 3.5.2.

## Statistical methods

For each tumor subtype, survival differences based on decades of diagnosis were also analyzed through Kaplan-Meier plots. We used an omnibus Mantel-Cox log-rank test to identify which Kaplan-Meier plots had significant changes in survival overtime and assessed significant trends with the logrank test for trend. In subtypes with significant omnibus tests, we identified the specific intervals when the significant change(s) occurred with post-hoc pairwise Mantel-Cox log-rank tests. Contiguous decades (ex 1970–1980) were first compared, followed by every two decades (1970–1990), and every three decades (1970-2000s). Additionally, we compared the survival curves of individual decades to all-decades of each subtype with pairwise Mantel-Cox log-rank test.

Holm-Sidak and Benjamini-Hochberg corrections were performed independently to correct for all multiple comparison tests. We reported the findings with Mantel-Haenszel hazard ratio, 95% confidence interval, and p-values of Mantel-Cox log-rank tests with the Holm-Sidak correction. The p-values of Mantel-Cox log-rank test with Benjamini-Hochberg correction were reported in supplementary tables. We defined statistical significance as p-values less than 0.05. One, five and ten year survival were calculated for each Kaplan-Meier plot and errors were calculated with exponential Greenwood formula. All statistical tests were performed through GraphPad Prism 6.0[15, 16].

## Results

### Participants and distribution of infantile brain tumors

We identified 2996 participants 24 months and under as having a diagnosis of a brain tumor between the years of 1973–2013. After grouping histological subtypes according to the 2016 World Health Organization classification of tumors in the CNS [17] (**Table 1**), diffuse astrocytic and oligodendroglial tumors were the most frequently diagnosed (30.7%), followed by pilocytic astrocytomas (18.5%), ependymal tumors (17.4%), embryonal tumors (17.0%), medulloblastoma (14.1%) and choroid plexus tumors (2.7%). Of note, the WHO classification scheme assigns medulloblastoma tumors to the category of "embryonal", however due to the high frequency of these tumors in the pediatric population, we categorized them individually.

### Survival is subtype dependent

Kaplan-Meier survival curves corresponding to the time period of 1973–2013 for each tumor subtype is shown in **Fig 1**. Infants with pilocytic astrocytoma had the longest survival

probability (95% CI) for 1, 5 and 10 years corresponding to 98% (97–99%), 94% (92–96%) and 93% (90–95%) (**S1 Table**). Infants with diffuse astrocytic and oligodendroglial tumors had a 1 year probability of survival >90%, however the 5 and 10 year survival probability was lower at ~ 80%. Similarly, infants with ependymal and choroid plexus tumors had a 1-year survival probability of 86–89% with a sharp decline to ≤ 65% in 5 year and 10 year survival probability. The shortest 1, 5, and 10 survival probability was observed in infants with medulloblastoma and embryonal tumors. Infants with medulloblastoma had a survival probability of 75% (70–79%) at 1 year, 59% (54–64%) at 5 years and 58% (52–63%) at 10 years. Infants with embryonal tumors had a survival probability of 63% (58–67%) at 1 year, 48% (43–53%) at 5 years, and 46% (41–51%) at 10 years.

All pairwise logrank comparisons of Kaplan-Meier survival curves between tumor subtypes were significant except for: diffuse astrocytic and oligodendroglial and choroid plexus; choroid plexus and ependymal; choroid plexus and medulloblastoma; and ependymal and medulloblastoma (**S2 Table**).

## The survival of infants with brain tumors has improved since the 1970s

We next set out to determine if the survival of infants with brain tumors has changed over time. First, we evaluated all tumors combined (**Fig 2**) and identified a positive trend in survival when comparing the past five decades (p = <0.0001). While the 1, 5, and 10-year survival improved with each decade from 1970s to 2010s, the only decade that experienced statically significant improvement in overall survival compared to the immediate prior decade was between 1980s and 1990s (Holm-Sidak corrected p = 0.032) (**S3** and **S4** Tables).

**Table 1. Most frequent tumor subtypes in infants.** Infantile brain tumors were classified into tumor subtypes (choroid plexus, diffuse astrocytic and oligodendroglial, embryonal, ependymal, medulloblastoma and pilocytic astrocytoma) based on histologies/ICD-O codes. The WHO 2016 Classification of Tumors of the CNS was used as a guide. The number of patients reported by SEER between the years of 1973–2013 is noted.

| Choroid Plexus | Number of Patients (n) |
|---|---|
| Choroid Plexus Carcinoma | 78 |
| **Diffuse Astrocytic and oligodendroglial** | **Number of Patients (n)** |
| Astrocytoma | 271 |
| Glioma | 398 |
| Glioblastoma | 60 |
| Anaplastic astrocytoma | 51 |
| Fibrillary astrocytoma | 39 |
| Oligodendroglioma | 35 |
| Mixed glioma | 28 |
| **Embryonal** | **Number of Patients (n)** |
| Primitive neuroectodermal | 227 |
| Atypical teratoid / rhabdoid | 168 |
| Neuroblastoma | 73 |
| Malignant rhabdoid | 19 |
| **Ependymal** | **Number of Patients (n)** |
| Ependymoma | 239 |
| Anaplastic ependymoma | 260 |
| **Medulloblastoma** | **Number of Patients (n)** |
| Medulloblastoma | 393 |
| **Pilocytic astrocytoma** | **Number of Patients (n)** |
| Pilocytic astrocytoma | 531 |

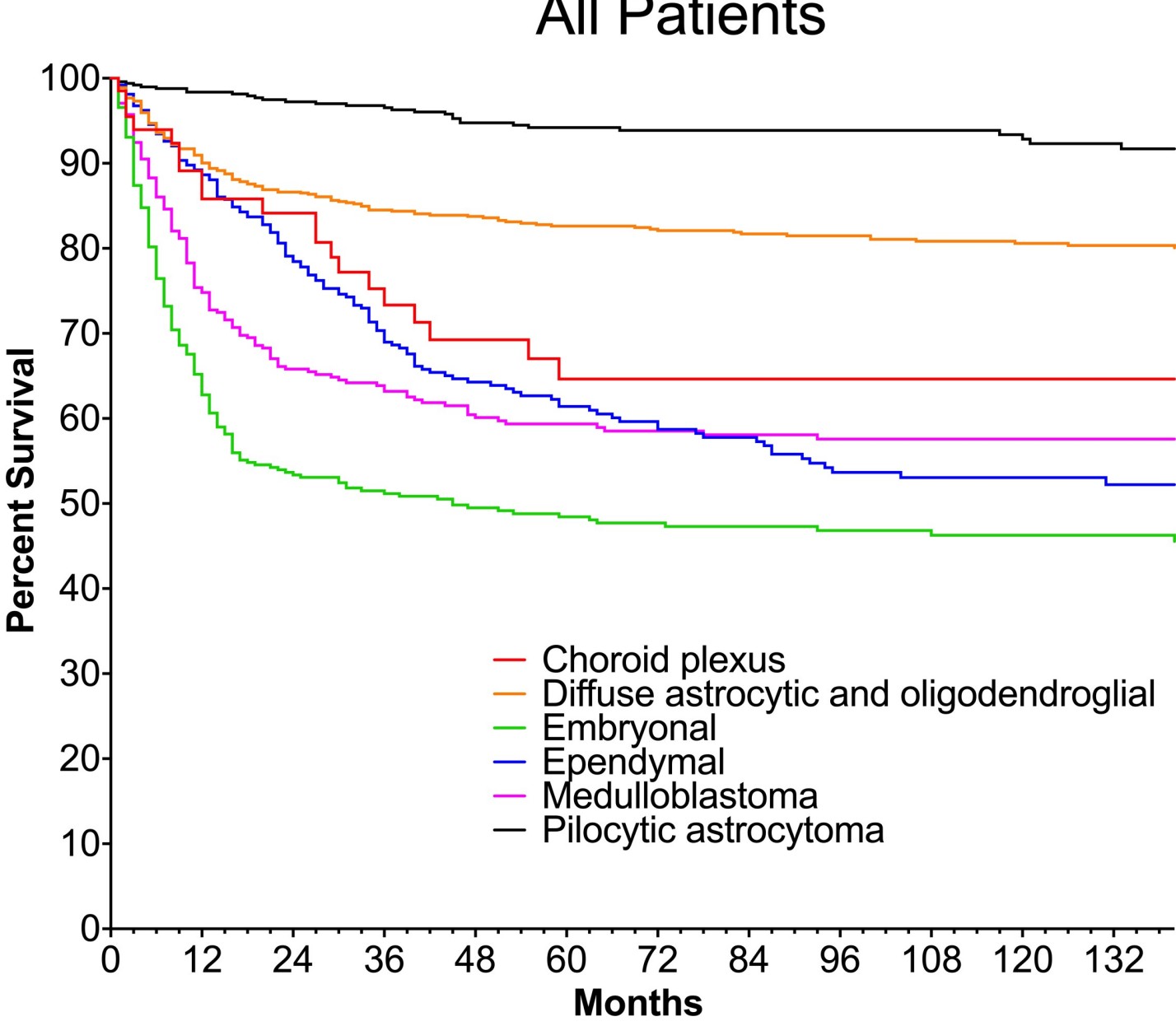

**Fig 1. Survival of infants with brain tumors by subtype.** Kaplan-Meier survival curves are shown with each color representing a different tumor subtype. Curves show the cumulative survival between the years of 1973–2013. Corresponding 1, 5, 10 year survivals are shown in S1 Table.

### Survival of subtypes by decade reveals differences in historical vs. contemporary survival curves

We subsequently investigated if the trend towards improved survival over time was also observed with respect to tumor subtypes. Kaplan-Meier survival curves (**Fig 3**) demonstrated that all subtypes except for embryonal tumors and choroid plexus tumors followed a statistically significant trend towards improved survival as a function of time. Choroid plexus showed no significant trend. Surprisingly, embryonal tumors showed a significant trend towards decreased survival as a function of time. The omnibus log-rank test demonstrated statistical

# All Patients

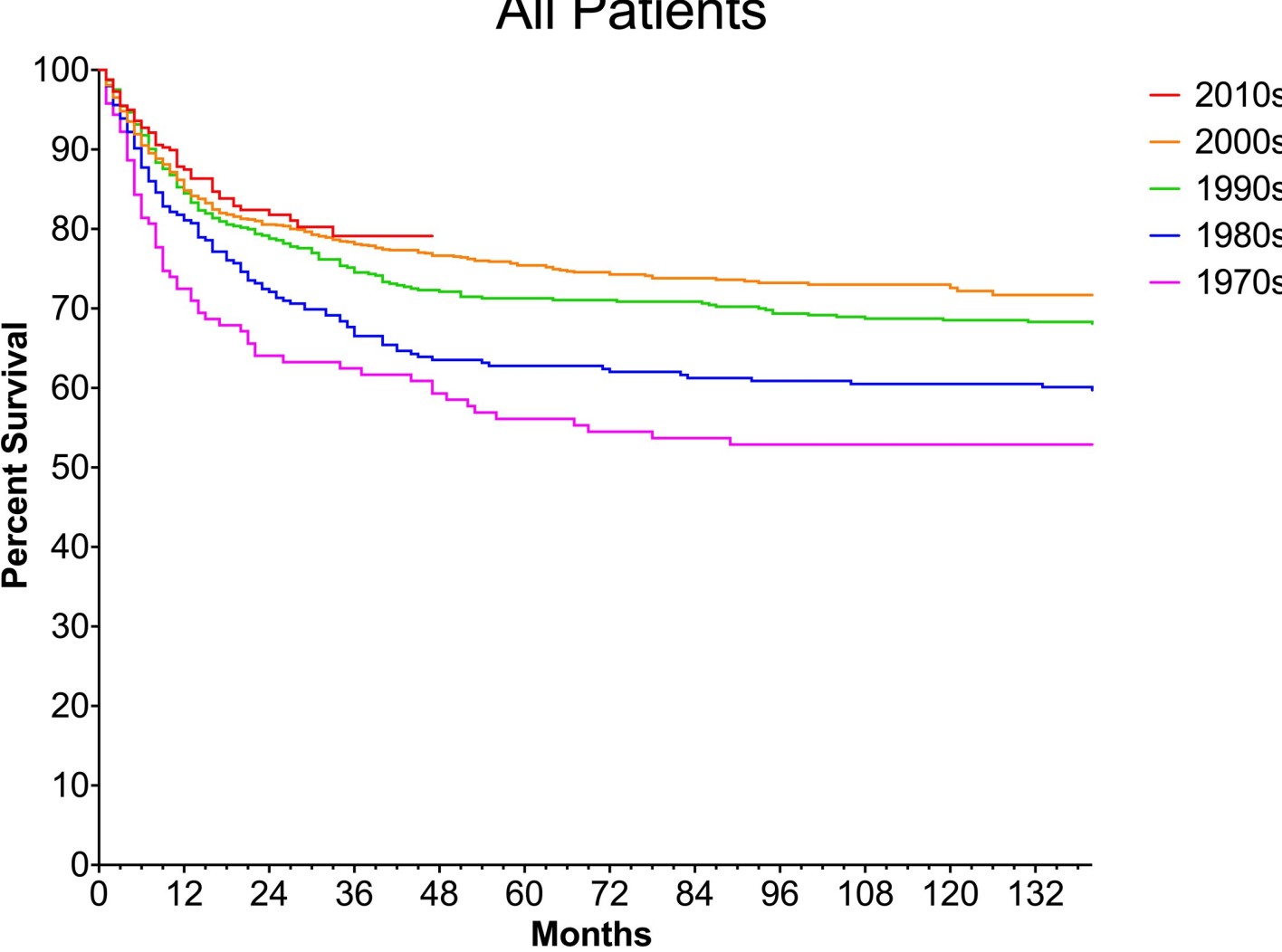

**Fig 2. Survival of infants with brain tumors by decade.** Kaplan-Meier survival curves are shown with each color representing a different decade. Corresponding 1, 5, 10 year survivals are shown in S3 Table.

significance between decades for all of the tumor types except for choroid plexus (S5 Table). All pairwise comparisons are summarized in S6 Table.

Statistically significant improvements in survival of infants was seen in pairwise comparisons between: 1970 and 1980 for diffuse astrocytic and oligodendroglial tumors (p = 0.0475; HR (Hazard$_{1980s}$/Hazard$_{1970s}$) = 0.50); 1980 and 1990 for pilocytic astrocytoma (p = 0.0003; HR (Hazard$_{1990s}$/Hazard$_{1980s}$) = 0.09) and ependymal tumors (p = 0.0004; HR (Hazard$_{1990s}$/Hazard$_{1980s}$) = 0.36); 1990 and 2000 for ependymal tumors (p = 0.044; HR (Hazard$_{2000s}$/Hazard$_{1990s}$) = 0.55); and 1980 and 2000 for medulloblastoma (p = 0.049; HR (Hazard$_{2000s}$/Hazard$_{1990s}$) = 0.61).

Statistically significant decline in survival of infants was only seen in pairwise comparison between 1970s and 1990s for embryonal tumors (p = 0.026; HR (Hazard$_{1990s}$/Hazard$_{1970s}$) = 2.81). The survival did not improve in subsequent decades.

The one, five, and ten-year survival are summarized in S7 Table.

We also compared the survival curves of each tumor subtype by individual decades to all-decades combined (Fig 3). The decades that were significantly worse than all-decades

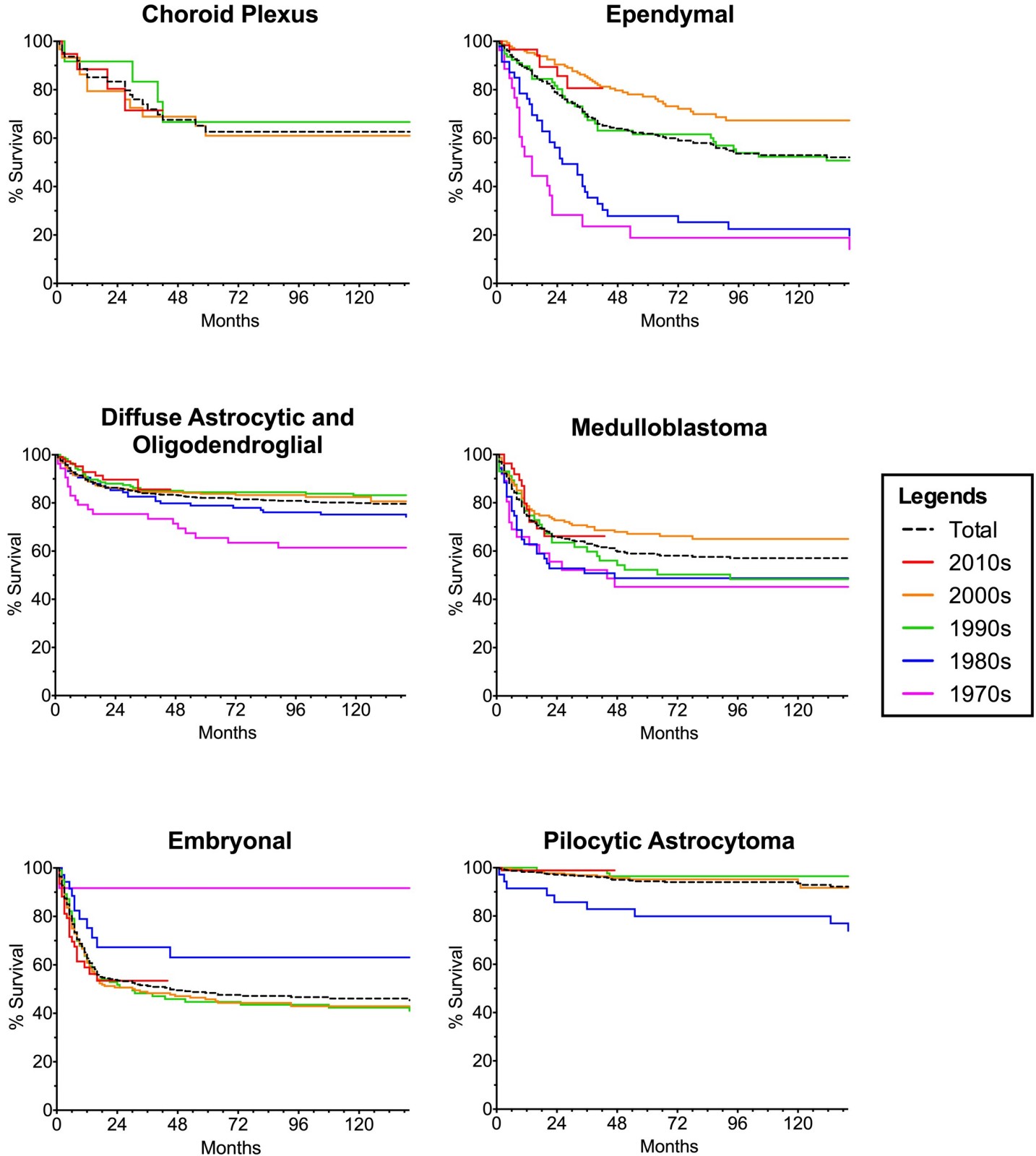

**Fig 3. Survival of infantile brain tumor subtypes by decade.** KM curves showing survival of each tumor subtype by decade. Each color represents a different decade, and the black dotted line represents average survival from 1973–2013. Corresponding 1, 5, and 10 year survivals are found in S7 Table.

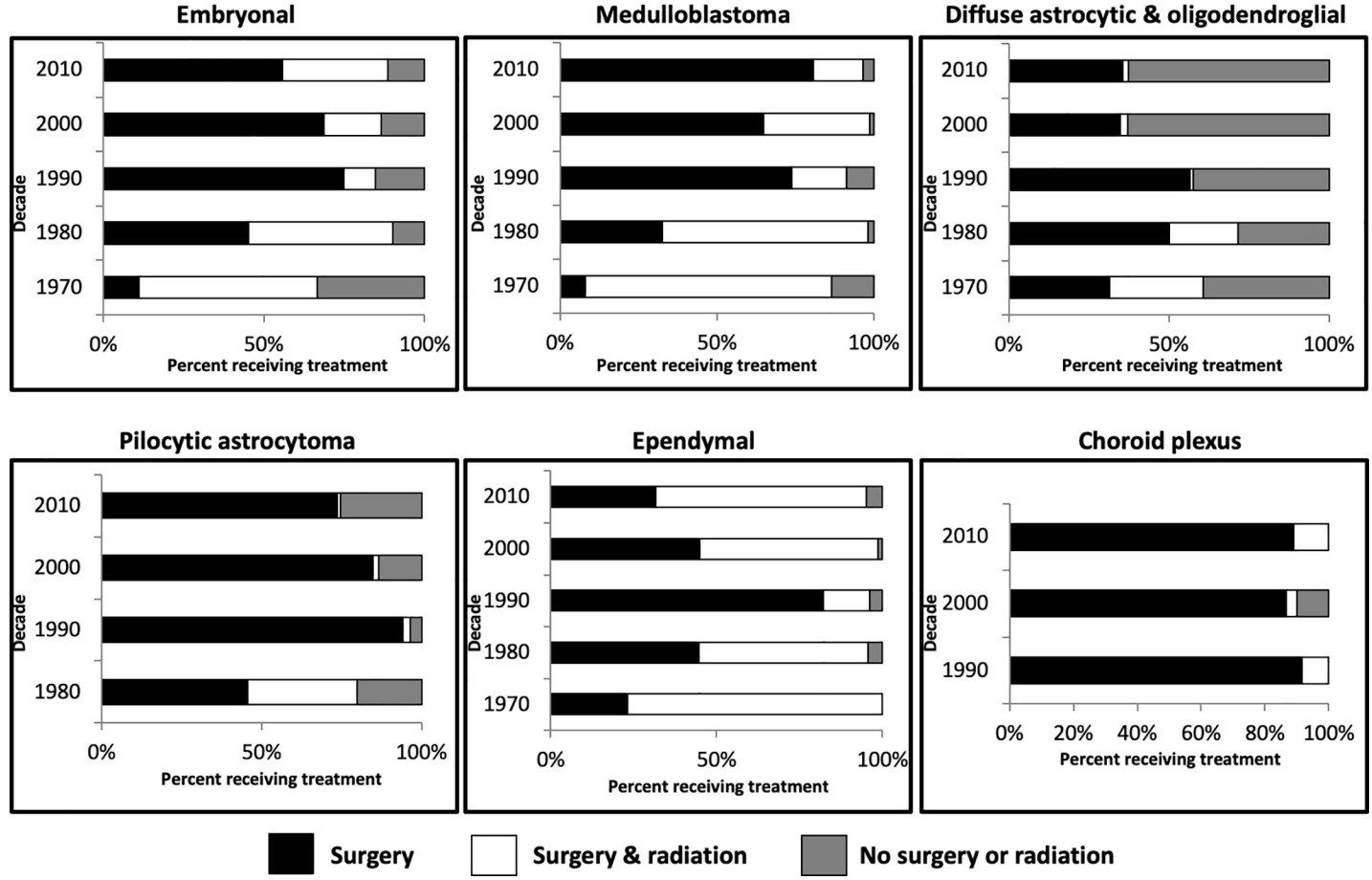

**Fig 4. Treatment by decade.** Bar graphs showing percentage of each treatment modality by decade. Black represents surgery, white (surgery & radiation), and dark grey (no surgery or radiation).

combined were: 1970s for diffuse astrocytic and oligodendroglial, 1980s for pilocytic astrocytoma, and 1970s and 1980s for ependymal tumors. The only decade that was significantly better than all-decades combined was 2000s for ependymal tumors (Holm-Sidak correct $p = 0.004$; HR ($HR_{2000s}$ /$HR_{All-Decades}$ = 0.6) (**S8 Table**).

### Treatment approaches to infantile brain tumors have changed over time

Given changes in infantile brain tumor survival over the past five decades, we sought to investigate if treatment approaches have changed over time (**Fig 4**). The SEER database provides data regarding surgery and radiation, but the use of chemotherapy or other adjuvant therapies is unknown. Accordingly, patients received "surgery only", "radiation only", combined "surgery & radiation" or "no surgery or radiation". Patients in the "no surgery or radiation" group had surgery (i.e. biopsy) for histological diagnosis of their tumors, however tumor resection/surgery with a therapeutic intent was not performed. There were no patients who received "radiation only".

### Surgery and radiation

All tumor subtypes had a decline in use of combined surgery & radiation in the 1990s compared earlier decades (1970s/1980s). Since the 1990s, the use of combined surgery & radiation has remained low (<3%) in infants with pilocytic astrocytomas and diffuse astrocytic and

oligodendrogliomas. However, treatment with combined surgery and radiation in contemporary cohorts (2000/2010s) has increased for infants with ependymal and embryonal tumors. In 2010s, approximately 64% of infants with ependymal tumors received combined surgery and radiation, compared to only 14% in 1990s. Likewise, for infants with embryonal tumors, 9.8% were treated with surgery and radiation in 1990s, which has increased to 32.8% in 2010s.

### Surgery only

For all tumor subtypes, there was a trend towards increased use of surgery only in the 1990s compared to the 1970s/1980s. In infants with medulloblastoma, treatment with surgery only has increased from 8.1% in 1970s to 80.7% in 2010s. Since the 1990s, surgery only is the most commonly used treatment modality for infants with choroid plexus tumors, medulloblastoma, embryonal, and pilocytic astrocytomas.

### No surgery or radiation

No resective surgery or radiation, which may represent biopsy followed by either no treatment or treatment with chemotherapy only, has increased for both diffuse astrocytic and oligodendroglial tumors and pilocytic astrocytomas. For infants with diffuse astrocytic and oligodendroglial tumors, treatment with "no surgery or radiation" increased from 29% to 63% and is the most common treatment starting in the 2000s. A smaller increase of 5% in "no surgery or radiation" approach was observed from 1980 to 2010s in infants with pilocytic astrocytomas.

## Discussion

As a group, the short and long-term survival of infants with brain tumors has been improving since the 1970s. Subtype analysis reveals this trend holds true for all subtypes except for infants with choroid plexus and embryonal tumors. For infants with choroid plexus tumors, the length of survival has not improved. For infants with embryonal tumors the length of survival has declined. The largest improvement in survival was observed in infants with ependymal tumors, which was the only group in which composite SEER data differed from contemporary survival curves. As such, for infants with ependymal tumors, up to date prognostication should be based on contemporary survival data. Treatment approaches have also changed for tumor subtypes over time. Treatment of all tumor subtypes except choroid plexus, experienced a decline in combined "surgery & radiation", and an increase in "surgery only" in the 1990s. Yet in contemporary cohorts "surgery & radiation" has increased as a treatment modality for ependymal and embryonal tumors, supporting the radio-sensitivity of these tumors.

### Embryonal tumors

Based on SEER analysis, we found that infants with embryonal tumors had the poorest survival. This finding is likely owing to the highly malignant nature of the most common histological subtypes in the group–supratentorial primitive neuroectodermal (PNET) and atypical teratoid/rhabdoid tumors (AT/RT). Analysis of the Austrian Brain Tumor Registry and the German HIT database found children with AT/RT between the ages of 0–14 years had a 5-year survival of 39.5% [18]. Likewise for PNET, analysis of children <19 years of age by the Canadian Pediatric Brain Tumor Consortium found a 4-year survival of 37.7% +/- 7.6%[19]. Upon analysis of survival by decade, we observed a sharp decline in survival of infants with embryonal tumors in the 1990s. This decline is most likely due to a combination of AT/RT being recognized as a distinct histological subtype in 1996, previously being misclassified as medulloblastoma[20] and decreased use of radiation in embryonal tumors starting in the

1990s. While the role of radiation is controversial in infants, radiotherapy has been associated with improved survival in infants with PNET and AT/RT. Based on the German HIT-SKK87 and HIT-SKK92 trials evaluating children <3 years with PNET, the 3 year progression-free survival was 24.1% in children who received radiation, compared to 6.7% in those who did not [21]. While prospective data is lacking for AT/RT, retrospective analysis of children at St. Jude's Children's Research Hospital[22] and Taipei Veteran's General Hospital in Taiwan [23] have observed long term survival almost exclusively in children receiving radiation. Perhaps in response to these studies, we observed a resurgence of combined radiation & surgery in the treatment of infants with embryonal tumors in the 2000s and 2010s. However, the survival benefit of radiation must be weighted against the numerous secondary effects of radiation such as cognitive decline, hormonal dysfunction, secondary malignancy, cerebrovascular disease and cranial neuropathy, which we were unable to assess in this study[24].

## Ependymal tumors

In our study, the most notable feature of ependymal tumors was the large improvement in survival over the past five decades. Changes in treatment approach may be a contributing factor. In a prospective study of 153 patients with ependymoma with a median age of 2.9 years, overall survival was not affected by treatment but was affected by tumor grade, extent of resection (GTR vs STR), sex and ethnic origin of the patient[25]. However, a recent retrospective analysis consisting of 360 cases from the SEER database, and 103 cases from two different institutions found that even with GTR, the long-term survival of children with ependymoma is poor with 10-year survival of 50% +/- 5% [26]. Similarly, using the SEER database we observed a 10-year survival of 53% +/-6%, however when analyzed by decade, the 10-year survival in the 2000s has improved to 67% +/-8.5%. This difference emphasizes the need to use contemporary datasets for the most up to date prognostication. Furthermore, while not pursued in this study, a more accurate prediction of survival of infants with ependymomas may be dependent on molecular subtype and tumor location (supratentorial vs. infratentorial vs. spine)[27].

## Medulloblastoma

For infants with medulloblastoma, we also observed a change in treatment approach over time, with a decline in combined surgery & radiation. This change in treatment is likely due to recognition of the neurocognitive effects associated with radiotherapy in young children, and clinical trial efforts in the 1990s to delay or avoid radiotherapy in this age group by using adjuvant chemotherapy[28–30]. In children under 3 years of age, chemotherapy alone after surgery was shown to be effective in children without metastases who had a gross-total resection[31]. While we did not analyze survival based on extent of surgical resection, we found no significant change in the 1-year survival since the 1970s. In contrast, long-term survival has improved since the 1970s, however contemporary 5 and 10-year survival remains poor at less than 67%. The poor survival may be due to few infants receiving a gross-total resection, or may reflect the aggressive biology of medulloblastoma in infants. Infants are predominately diagnosed with Sonic hedgehog (SHH) or group 3 histological subgroups[32], with 40–50% of the group 3 tumors having metastases at diagnosis[33]. A retrospective study of 53 patients with a median age of 24 months found a 5-year overall survival of 86.2% +/- 7.4 in the SHH group, and only 49.1 +/- 14% in infants with group 3 tumors[34].

## Pilocytic astrocytoma

Analysis of survival by decade identified a statistically significant improvement in survival from the 1980s to the 1990s. Treatment approach also changed during these two decades with

34% of infants receiving surgery/radiation in the 1980s, compared to only 3.5% in the 1990s. The decline in combined surgery & radiation was followed by an increase in surgery only. This change reflects the clinical observations that complete surgical resection of pilocytic astrocytomas is usually curative, and radiation has not been shown to extend survival in patients with incomplete resection in clinical trials[35]. Hence the improvement in survival observed between the 1980s and contemporary cohorts (1990s-2010s), may be due to improvements in microsurgical technique[36]. The growing use of chemotherapy may also be a contributor of improving survival especially in children with non-resectable disease[37].

### Diffuse astrocytic and oligodendroglial tumors

Evaluation of survival by decade showed a significant improvement in 1, 5, and 10 year between the years of 1970s and 1980s. During these decades we observed an increase in surgery only treatment with a decline in combined surgery & radiation. However, we cannot determine if this treatment change was responsible for the improved survival. By the 1990s to 2010s, we observed that the use of combined surgery & radiation was almost non-existent, and had been replaced by an increase in "no surgery or radiation". This change in therapy may be due to the development of more effective chemotherapeutic and neoadjuvant therapies, as well as the recognition of the often infiltrating nature of these tumors [37–39]. In comparison to older children, there remains no standard of care treatment, and there are few studies comparing treatment modalities in infantile gliomas and astrocytomas [40].

### Choroid plexus tumors

SEER data available for this report combined all histological subtypes of choroid plexus tumors, but given that SEER did not include benign tumors (i.e. choroid plexus papilloma) until 2004 and WHO did not recognized atypical choroid plexus papilloma until 2007, all the of the SEER choroid plexus tumors diagnosed up to 2004 were choroid plexus carcinomas. Unlike other tumor subgroups, we observed no significant changes in survival over time. A previous SEER report exclusively on choroid plexus tumors in children <20 years of age diagnosed from 1978–2010 demonstrated 98% survival for children with choroid plexus papilloma with a 0% tumor specific mortality rate, which suggests that the poor survival in this infant SEER report is predominately due to choroid plexus carcinoma[41]. Similar to our findings, the group from Johns Hopkins reported an average 5-year survival for children with choroid plexus carcinoma of 71%, with a median age at diagnosis being 3 years[42]. However, multiple other groups have reported 5-year survivals <50% [43–45]. It is unclear why there is a large discrepancy in survival between studies, but treatment may be a contributing factor. In our study we observed that the treatment of choroid plexus tumors has remained stable since the 1990s, with surgery being the predominant approach. The extent of resection has been found to be the most important prognostic factor[43], and studies with poor survival noted that not all children were candidates for gross-total resection[44]. Another possible reason for large variations in reported survival may be due to combining survival data of choroid plexus carcinoma with choroid plexus papilloma. The 5-year survival of children with choroid plexus papilloma is 100% compared to 71% survival in children with choroid plexus carcinoma[42]. In our study, we identified 78 children with choroid plexus tumors, of which 25 were defined as carcinoma with the rest classified as "unknown". This unknown group may represent carcinoma or papilloma since the ICD codes are the same for the histological entities.

### Limitations

Limitations of the SEER database have been described by our group[14], and there are several additional limitations specific to this study. As previously mentioned, treatment was

categorized into surgery, surgery & radiation, and no surgery or radiation. We interpreted that the "no surgery or radiation" represents no resection or radiation, however the SEER database provides no specification. Additionally, for multiple subtypes (ependymal tumors, medullo-blastoma, pilocytic astrocytomas and choroid plexus tumors), the extent of resection has been reported to be associated with survival. While the SEER database provides some codes includ-ing "resection of lobe of brain" and "subtotal resection", these codes are not standardized or provided for all patients. Also, while the SEER database has a large sample size, the recorded outcomes (i.e. survival/death) are limited. Outcomes such as quality of life, and cognitive func-tion are not included and are of particular importance in the pediatric population.

## Conclusion

The prognosis of infants with brain tumors has been unclear due to the rarity of these tumors. Using a population-based approach we provide up-to-date prognostication of infant brain tumors by subtype. We compared contemporary and historical survival and found that for the majority of subtypes survival has improved over time. Treatment approaches have also evolved, with a decline in radiation use over time, but we also showed a recent increase in use for radiosensitive cancers. This study provides a basis for subsequent investigations addressing how specific treatments (surgery vs. radiation vs. chemotherapy) have altered infant survival. This study also highlights the need to identify new treatment modalities for infants with tumors that continue to have a dismal prognosis, such as embryonal tumors.

## Supporting information

**S1 Table. One, five and ten-year survival probabilities for infants with the most common histological subtypes of tumors (choroid plexus, diffuse astrocytic and oligodendroglial, embryonal, ependymal, pilocytic astrocytoma and medulloblastoma).**
(XLSX)

**S2 Table. Comparison of Kaplan Meier survival curves between different tumor subtypes using pairwise log rank tests with Holm-Sidak and Benjamini-Hochberg corrections.**
(XLSX)

**S3 Table. Table showing the 1-, 5-, and 10-year survival of infants with brain tumors by decade.**
(XLSX)

**S4 Table. Comparison of Kaplan Meier survival curves between different decades of diag-nosis of all patients using pairwise comparisons with Holm-Sidak and Benjamini-Hoch-berg corrections.**
(XLSX)

**S5 Table. Omnibus log-rank test comparing survival between different decades.**
(XLSX)

**S6 Table. Comparison of Kaplan Meier survival curves between different decades of diag-nosis within each tumor subtype using pairwise comparisons with Holm-Sidak and Benja-mini-Hochberg corrections.**
(XLSX)

**S7 Table. The 1-, 5- and 10-year survival of different tumor subtypes by decade.**
(XLSX)

**S8 Table. Comparison of Kaplan Meier survival curves for each decade to all-decades combined using pairwise log-rank tests with Holm-Sidak and Benjamini-Hochberg corrections.**
(XLSX)

## Author Contributions

**Conceptualization:** George Ibrahim, Anthony Wang, Alexander Weil, Aria Fallah, Albert Tu.

**Data curation:** Timothy Chai, Sharjeel Syed, Lior Elkaim, Aria Fallah.

**Formal analysis:** Timothy Chai, Sharjeel Syed, Nathan Lau, Lior Elkaim, Anne Bendel, Aria Fallah, Albert Tu.

**Investigation:** Anne Bendel, Aria Fallah.

**Methodology:** Anne Bendel, Aria Fallah, Albert Tu.

**Project administration:** Aria Fallah.

**Supervision:** Alexander Weil, Aria Fallah, Albert Tu.

**Writing – original draft:** Claire Faltermeier, Timothy Chai, Sharjeel Syed, Lior Elkaim, George Ibrahim, Anne Bendel.

**Writing – review & editing:** Claire Faltermeier, Timothy Chai, Lior Elkaim, George Ibrahim, Anthony Wang, Alexander Weil, Anne Bendel, Aria Fallah, Albert Tu.

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
