## [Decision Letter · Decision Letter 0]

9 Sep 2019

[EXSCINDED]

PONE-D-19-21460

Survival of Infants ≤24 months of age with brain tumors: a population-based study using the SEER database

PLOS ONE

Dear Dr Tu,

Thank you for submitting your manuscript to PLOS ONE. After careful consideration, we feel that it has merit but does not fully meet PLOS ONE’s publication criteria as it currently stands. Therefore, we invite you to submit a revised version of the manuscript that addresses the points raised during the review process.

The authors distinguish PNET and medulloblastoma.  I would imagine that PNET refers to supratentorial PNET? Please clarify.

We would appreciate receiving your revised manuscript by Oct 24 2019 11:59PM. To enhance the reproducibility of your results, we recommend that if applicable you deposit your laboratory protocols in protocols.io, where a protocol can be assigned its own identifier (DOI) such that it can be cited independently in the future. For instructions see: http://journals.plos.org/plosone/s/submission-guidelines#loc-laboratory-protocols

We look forward to receiving your revised manuscript.

Kind regards,

Jonathan H Sherman

Academic Editor

PLOS ONE

Journal Requirements:

1. Please include your tables as part of your main manuscript and remove the individual files. Please note that supplementary tables (should remain/ be uploaded) as separate "supporting information" files

2. Please amend your list of authors on the manuscript to ensure that each author is linked to an affiliation. Authors’ affiliations should reflect the institution where the work was done (if authors moved subsequently, you can also list the new affiliation stating “current affiliation:….” as necessary).

Reviewers' comments:

Reviewer's Responses to Questions

**Comments to the Author**

1. Is the manuscript technically sound, and do the data support the conclusions?

Reviewer #1: Yes

Reviewer #2: Yes

2. Has the statistical analysis been performed appropriately and rigorously? 

Reviewer #1: Yes

Reviewer #2: Yes

3. Have the authors made all data underlying the findings in their manuscript fully available?

Reviewer #1: Yes

Reviewer #2: Yes

4. Is the manuscript presented in an intelligible fashion and written in standard English?

Reviewer #1: Yes

Reviewer #2: Yes

5. Review Comments to the Author

Reviewer #1: Feltermeier et al reviewed the files of 2996 affected children, aged 24 months and younger, with brain tumors from the Surveillance, Epidemiology, and End Result (SEER) database between 1973 and 2013. The authors concluded that overall survival for infants with brain tumors has improved from the 1970’s, with the exception of choroid plexus and embryonal tumors. They present data on “1) distribution of infantile brain tumors, 2) survival strings for brain tumors by subtype and epoch, and 3) trends in treatment and shifts in therapeutic approaches over time.” The authors cite their prior work describing the SEER database and add some intrinsic limitations with the SEER database for these specific questions, most importantly the quantification of extent of resection and granular data on outcomes, especially important in pediatrics. Overall this is very carefully thought out work that adds to the literature by stratifying this rare patient population by histological diagnosis, addressing the important improvements in outcomes, correlating with changes of care over time.

Reviewer #2: In this study, the authors used the SEER database to analyze treatment and survival of infants (less than 24 months of age) from 1973-2013. They identified 2996 patients over this time, and analyzed by brain tumor type, decade and treatment. Within the limits of this database, they found improved survival overall, although some tumor types did not show improved survival. In addition, there was decreased use of radiation, except for embryonal tumors. I commend the authors on this large study. I can imagine it would be quite difficult to do this analysis, especially with ever-changing terminology over time. Some of the results can be expected, but this type of analysis is extremely important to have a better overall understanding and perspective of brain tumor treatment over time. In addition, as the authors point out, we need to use contemporary data in doing studies, as outcomes have changed over time. One minor point, they distinguish PNET and medulloblastoma. I would imagine that PNET refers to supratentorial PNET? I would just ask that they clarify. Nice paper.

6. PLOS authors have the option to publish the peer review history of their article (what does this mean?). If published, this will include your full peer review and any attached files.

Reviewer #1: Yes: Carlos E. Sanchez, MD

Reviewer #2: No

---

## [Author Response · Author response to Decision Letter 0]

11 Sep 2019

Thank you for the reviewer comments. We have clarified in the body of the manuscript that PNET refers to supratentorial PNET.

Table 1 has now been moved into the body of the manuscript.

---

## [Editor Report · Decision Letter 1]

13 Sep 2019

Survival of Infants ≤24 months of age with brain tumors: a population-based study using the SEER database

PONE-D-19-21460R1

Dear Dr. Tu,

We are pleased to inform you that your manuscript has been judged scientifically suitable for publication and will be formally accepted for publication once it complies with all outstanding technical requirements.

With kind regards,

Jonathan H Sherman

Academic Editor

PLOS ONE
---

## [Editor Report · Acceptance letter]

18 Sep 2019

PONE-D-19-21460R1 

Survival of Infants ≤24 months of age with brain tumors: a population-based study using the SEER database 

Dear Dr. Tu:

I am pleased to inform you that your manuscript has been deemed suitable for publication in PLOS ONE. Congratulations! Your manuscript is now with our production department. 

With kind regards,

on behalf of

Dr. Jonathan H Sherman 

Academic Editor

PLOS ONE